# NEURAL TIME-DEPENDENT PARTIAL DIFFERENTIAL EQUATION

## ABSTRACT

Partial differential equations (PDEs) play a crucial role in studying a vast number of problems in science and engineering. Numerically solving nonlinear and/or high-dimensional PDEs is frequently a challenging task. Inspired by the traditional finite difference and finite elements methods and emerging advancements in machine learning, we propose a sequence-to-sequence learning (Seq2Seq) framework called Neural-PDE, which allows one to automatically learn governing rules of any time-dependent PDE system from existing data by using a bidirectional LSTM encoder, and predict the solutions in next $n$ time steps. One critical feature of our proposed framework is that the Neural-PDE is able to simultaneously learn and simulate all variables of interest in a PDE system. We test the Neural-PDE by a range of examples, from one-dimensional PDEs to a multi-dimensional and nonlinear complex fluids model. The results show that the Neural-PDE is capable of learning the initial conditions, boundary conditions and differential operators defining the initial-boundary-value problem of a PDE system without the knowledge of the specific form of the PDE system. In our experiments, the Neural-PDE can efficiently extract the dynamics within 20 epochs training and produce accurate predictions. Furthermore, unlike the traditional machine learning approaches for learning PDEs, such as CNN and MLP, which require great quantity of parameters for model precision, the Neural-PDE shares parameters among all time steps, and thus considerably reduces computational complexity and leads to a fast learning algorithm.

## 1 INTRODUCTION

The research of time-dependent partial differential equations (PDEs) is regarded as one of the most important disciplines in applied mathematics. PDEs appear ubiquitously in a broad spectrum of fields including physics, biology, chemistry, and finance, to name a few. Despite their fundamental importance, most PDEs can not be solved analytically and have to rely on numerical solving methods. Developing efficient and accurate numerical schemes for solving PDEs, therefore, has been an active research area over the past few decades (Courant et al., 1967; Osher & Sethian, 1988; LeVeque; Cockburn et al., 2012; Thomas, 2013; Johnson, 2012). Still, devising stable and accurate schemes with acceptable computational cost is a difficult task, especially when nonlinear and(or) high-dimensional PDEs are considered. Additionally, PDE models emerged from science and engineering disciplines usually require huge empirical data for model calibration and validation, and determining the multi-dimensional parameters in such a PDE system poses another challenge (Peng et al., 2020).

Deep learning is considered to be the state-of-the-art tool in classification and prediction of nonlinear inputs, such as image, text, and speech (Litjens et al., 2017; Devlin et al., 2018; LeCun et al., 1998; Krizhevsky et al., 2012; Hinton et al., 2012). Recently, considerable efforts have been made to employ deep learning tools in designing data-driven methods for solving PDEs (Han et al., 2018; Long et al., 2018; Sirignano & Spiliopoulos, 2018; Raissi et al., 2019). Most of these approaches are based on fully-connected neural networks (FCNNs), convolutional neural networks(CNNs) and multilayer perceptron (MLP). These neural network structures usually require an increment of the layers to improve the predictive accuracy (Raissi et al., 2019), and subsequently lead to a more complicated model due to the additional parameters. Recurrent neural networks (RNNs) are one type of neural network architectures. RNNs predict the next time step value by using the input data from the current

and previous states and share parameters across all inputs. This idea (Sherstinsky, 2020) of using current and previous step states to calculate the state at the next time step is not unique to RNNs. In fact, it is ubiquitously used in numerical PDEs. Almost all time-stepping numerical methods applied to solve time-dependent PDEs, such as Euler's, Crank-Nicolson, high-order Taylor and its variance Runge-Kutta (Ascher et al., 1997) time-stepping methods, update numerical solution by utilizing solution from previous steps.

This motivates us to think what would happen if we replace the previous step data in the neural network with numerical solution data to PDE supported on grids. It is possible that the neural network behaves like a time-stepping method, for example, forward Euler's method yields the numerical solution at a new time point as the current state output (Chen et al., 2018). Since the numerical solution on each of the grid point (for finite difference) or grid cell (for finite element) computed at a set of contiguous time points can be treated as neural network input in the form of one time sequence of data, the deep learning framework can be trained to predict any time-dependent PDEs from the time series data supported on some grids if the bidirectional structure is applied (Huang et al., 2015; Schuster & Paliwal, 1997). In other words, the supervised training process can be regarded as a practice of the deep learning framework to learn the numerical solution from the input data, by learning the coefficients on neural network layers.

Long Short-Term Memory (LSTM) (Hochreiter & Schmidhuber, 1997) is a neural network built upon RNNs. Unlike vanilla RNNs, which suffer from losing long term information and high probability of gradient vanishing or exploding, LSTM has a specifically designed memory cell with a set of new gates such as input gate and forget gate. Equipped with these new gates which control the time to preserve and pass the information, LSTM is capable of learning long term dependencies without the danger of having gradient vanishing or exploding. In the past two decades, LSTM has been widely used in the field of natural language processing (NLP), such as machine translation, dialogue systems, question answering systems (Lipton et al., 2015).

Inspired by numerical PDE schemes and LSTM neural network, we propose a new deep learning framework, denoted as Neural-PDE. It simulates multi-dimensional governing laws, represented by time-dependent PDEs, from time series data generated on some grids and predicts the next $n$ time steps data. The Neural-PDE is capable of intelligently processing related data from all spatial grids by using the bidirectional (Schuster & Paliwal, 1997) neural network, and thus guarantees the accuracy of the numerical solution and the feasibility in learning any time-dependent PDEs. The detailed structures of the Neural-PDE and data normalization are introduced in Section 3.

The rest of the paper is organized as follows. Section 2 briefly reviews finite difference method for solving PDEs. Section 3 contains detailed description of designing the Neural-PDE. In Section 4 and Appendix A of the paper, we apply the Neural-PDE to solve four different PDEs, including the 1-dimensional(1D) wave equation, the 2-dimensional(2D) heat equation, and two systems of PDEs: the invicid Burgers' equations and a coupled Navier Stokes-Cahn Hilliard equations, which widely appear in multiscale modeling of complex fluid systems. We demonstrate the robustness of the Neural-PDE, which achieves convergence within 20 epochs with an admissible mean squared error, even when we add Gaussian noise in the input data.

## 2 PRELIMINARIES

### 2.1 TIME DEPENDENT PARTIAL DIFFERENTIAL EQUATIONS

A time-dependent partial differential equation is an equation of the form:

$$u_t = f(x_1, \cdots, x_n, u, \frac{\partial u}{\partial x_1}, \cdots, \frac{\partial u}{\partial x_n}, \frac{\partial^2 u}{\partial x_1 \partial x_1}, \cdots, \frac{\partial^2 u}{\partial x_1 \partial x_n}, \cdots, \frac{\partial^n u}{\partial x_1 \cdots \partial x_n}), \qquad (2.1.1)$$

where $u = u(t, x_1, ..., x_n)$ is known, $x_i \in \mathbb{R}$ are spatial variables, and the operator $f$ maps $\mathbb{R} \mapsto \mathbb{R}$. For example, consider the parabolic heat equation: $u_t = \alpha^2 \Delta u$, where $u$ represents the temperature and $f$ is the Laplacian operator $\Delta$. Eq. (2.1.1) can be solved by finite difference methods, which is briefly reviewed below for the self-completeness of the paper.

## 2.2 Finite Difference Method

Consider using a finite difference method (FDM) to solve a two-dimensional second-order PDE of the form:

$$u_t = f(x, y, u_x, u_y, u_{xx}, u_{yy}), \quad (x,y) \in \Omega \subset \mathbb{R}^2, \quad t \in \mathbb{R}^+ \cup \{0\}, \tag{2.2.1}$$

with some proper boundary conditions. Let $\Omega$ be $\Omega = [x_a, x_b] \times [y_a, y_b]$, and

$$u_{i,j}^n = u(x_i, y_j, t_n) \tag{2.2.2}$$

where $t_n = n\delta t$, $0 \le n \le N$, and $\delta t = \frac{T}{N}$ for $t \in [0, T]$, and some large integer $N$. $x_i = i\delta x$, $0 \le i \le N_x$, $\delta x = \frac{x_a - x_b}{N_x}$ for $x \in [x_a, x_b]$. $y_j = j\delta y$, $0 \le j \le N_y$, $\delta y = \frac{y_a - y_b}{N_y}$ for $y \in [y_a, y_b]$. $N_x$ and $N_y$ are integers.

The central difference method approximates the spatial derivatives as follows (Thomas, 2013):

$$u_x(x_i, y_j, t) = \frac{1}{2\delta x}(u_{i+1,j} - u_{i-1,j}) + \mathcal{O}(\delta x^2), \tag{2.2.3}$$

$$u_y(x_i, y_j, t) = \frac{1}{2\delta y}(u_{i,j+1} - u_{i,j-1}) + \mathcal{O}(\delta y^2), \tag{2.2.4}$$

$$u_{xx}(x_i, y_j, t) = \frac{1}{\delta x^2}(u_{i+1,j} - 2u_{i,j} + u_{i-1,j}) + \mathcal{O}(\delta x^2), \tag{2.2.5}$$

$$u_{yy}(x_i, y_j, t) = \frac{1}{\delta y^2}(u_{i,j+1} - 2u_{i,j} + u_{i,j-1}) + \mathcal{O}(\delta y^2). \tag{2.2.6}$$

To this end, the explicit time-stepping scheme to update next step solution $u^{n+1}$ is given by:

$$u_{i,j}^n \approx U_{i,j}^{n+1} = U_{i,j}^n + \delta t f(x_i, y_j, U_{i,j}^n, U_{i,j-1}^n, U_{i,j+1}^n, U_{i+1,j}^n, U_{i-1,j}^n), \tag{2.2.7}$$

$$\equiv \mathbf{F}(x_i, y_j, \delta x, \delta y, \delta t, U_{i,j}^n, U_{i,j-1}^n, U_{i,j+1}^n, U_{i+1,j}^n, U_{i-1,j}^n), \tag{2.2.8}$$

Apparently, the finite difference method (2.2.7) for updating $u^{n+1}$ on a grid point relies on the previous time steps' solutions, supported on the grid point and its neighbours. The scheme (2.2.7) updates $u_{i,j}^{n+1}$ using four points of $u^n$ values (see Figure 1). Similarly, the finite element method (FEM) approximates the new solution by calculating the corresponded mesh cell coefficient (see Appendix), which is updated by its related nearby coefficients on the mesh. From this perspective, one may regard the numerical schemes for solving time-dependent PDEs as methods catching the information from neighbourhood data of interest.

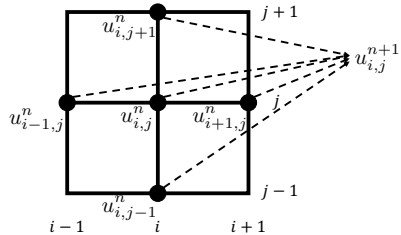

Figure 1: updating scheme for central difference method

## 3 Proposed Method

### 3.1 Mathematical Motivation

Recurrent neural network including LSTM is an artificial neural network structure of the form (Lipton et al., 2015):

$$\boldsymbol{h}^t = \sigma(\mathbf{W}^{hx}\boldsymbol{x}^t + \mathbf{W}^{hh}\boldsymbol{h}^{t-1} + \boldsymbol{b}_h) \equiv \sigma_a(\boldsymbol{x}^t, \boldsymbol{h}^{t-1}) \equiv \sigma_b(\boldsymbol{x}^0, \boldsymbol{x}^1, \boldsymbol{x}^2, \cdots, \boldsymbol{x}^t), \tag{3.1.1}$$

where $\boldsymbol{x}^t \in \mathbb{R}^d$ is the input data of the $t^{th}$ state and $\boldsymbol{h}^{t-1} \in \mathbb{R}^h$ denotes the processed value in its previous state by the hidden layers. The output $\boldsymbol{y}^t$ of the current state is updated by the current state value $\boldsymbol{h}^t$:

$$\boldsymbol{y}^t = \sigma(\mathbf{W}^{hy}\boldsymbol{h}^t + \boldsymbol{b}_y) \tag{3.1.2}$$

$$\equiv \sigma_c(\boldsymbol{h}^t) \equiv \sigma_d(\boldsymbol{x}^0, \boldsymbol{x}^1, \boldsymbol{x}^2, \cdots, \boldsymbol{x}^t). \tag{3.1.3}$$

Here $\mathbf{W}^{hx} \in \mathbb{R}^{h \times d}$, $\mathbf{W}^{hh} \in \mathbb{R}^{h \times h}$, $\mathbf{W}^{hy} \in \mathbb{R}^{h \times h}$ are the matrix of weights, vectors $\boldsymbol{b}_h, \boldsymbol{b}_y \in \mathbb{R}^h$ are the coefficients of bias, and $\sigma, \sigma_a, \sigma_b, \sigma_c, \sigma_d$ are corresponded activation and mapping functions.

With proper design of input and forget gate, LSTM can effectively yield a better control over the gradient flow and better preserve useful information from long-range dependencies (Graves & Schmidhuber, 2005).

Now consider a temporally continuous vector function $\boldsymbol{u} \in \mathbb{R}^n$ given by an ordinary differential equation with the form:

$$\frac{d\boldsymbol{u}(t)}{dt} = g(\boldsymbol{u}(t)) . \tag{3.1.4}$$

Let $\boldsymbol{u}^n = \boldsymbol{u}(t = n\delta t)$, a forward Euler's method for solving $\boldsymbol{u}$ can be easily derived from the Taylor's theorem which gives the following first-order accurate approximation of the time derivative:

$$\frac{d\boldsymbol{u}^n}{dt} = \frac{\boldsymbol{u}^{n+1} - \boldsymbol{u}^n}{\delta t} + \mathcal{O}(\delta t) . \tag{3.1.5}$$

Then we have:

$$\frac{d\boldsymbol{u}}{dt} = g(\boldsymbol{u}) \xrightarrow{(3.1.5)} \boldsymbol{u}^{n+1} = \boldsymbol{u}^n + \delta t \, g(\boldsymbol{u}^n) + \mathcal{O}(\delta t^2)$$

$$\rightarrow \hat{\boldsymbol{u}}^{n+1} = f_1(\hat{\boldsymbol{u}}^n) = \underbrace{f_1 \circ f_1 \circ \cdots f_1(\hat{\boldsymbol{u}}^0)}_{n} \tag{3.1.6}$$

Here $\hat{\boldsymbol{u}}^n \approx \boldsymbol{u}(n\delta t)$ is the numerical approximation and $f_1 \equiv \boldsymbol{u}^n + \delta t \, g(\boldsymbol{u}^n) : \mathbb{R}^n \to \mathbb{R}^n$. Combining equations (3.1.1) and (3.1.6) one may notice that the residual networks, recurrent neural network and also LSTM networks can be regarded as a numerical scheme for solving time-dependent differential equations if more layers are added and smaller time steps are taken. (Chen et al., 2018)

Canonical structure for such recurrent neural network usually calculate the current state value by its previous time step value $\boldsymbol{h}^{t-1}$ and current state input $\boldsymbol{x}^t$. Similarly, in numerical PDEs, the next step data at a grid point is updated from the previous (and current) values on its nearby grid points (see Eq. 2.2.7).

Thus, what if we replace the temporal input $\boldsymbol{h}^{t-1}$ and $\boldsymbol{x}^t$ with spatial information? A simple sketch of the upwinding method for a $1d$ example of $u(x,t)$:

$$u_t + \nu u_x = 0 \tag{3.1.7}$$

will be:

$$u_i^{n+1} = u_i^n - \nu \frac{\delta t}{\delta x}(u_i^n - u_{i-1}^n) + \mathcal{O}(\delta x, \delta t) \rightarrow \hat{u}_i^{n+1} = f_2(\hat{u}_{i-1}^n, \hat{u}_i^n) \tag{3.1.8}$$

$$\equiv f_\theta\big(f_\eta(\boldsymbol{x}_i, \boldsymbol{h}_{i-1}(u))\big) = f_{\theta,\eta}\big(\hat{u}_0^n, \hat{u}_1^n, \cdots, \hat{u}_{i-1}^n, \hat{u}_i^n\big) = v_i^{n+1} \tag{3.1.9}$$

$$\boldsymbol{x}_i = \hat{u}_i^n, \ \boldsymbol{h}_{i-1}(\hat{u}) = \sigma(\hat{u}_{i-1}^n, \boldsymbol{h}_{i-2}(\hat{u})) \equiv f_\eta(\hat{u}_0^n, \hat{u}_1^n, \hat{u}_2^n, \cdots, \hat{u}_{i-1}^n). \tag{3.1.10}$$

Here let $v_i^{n+1}$ be the prediction of $\hat{u}_i^{n+1}$ processed by neural network. We replace the temporal previous state $\boldsymbol{h}^{t-1}$ with spacial grid value $\boldsymbol{h}_{i-1}$ and input the numerical solution $\hat{u}_i^n \approx u(i\delta x, n\delta t)$ as current state value, which indicates the neural network could be seen as a forward Euler method for equation 3.1.7 (Lu et al., 2018). Function $f_2 \equiv \hat{u}_i^n - \nu \frac{\delta t}{\delta x}(\hat{u}_i^n - \hat{u}_{i-1}^n) : \mathbb{R} \to \mathbb{R}$ and the function $f_\theta$ represents the dynamics of the hidden layers in decoder with parameters $\theta$, and $f_\eta$ specifies the dynamics of the LSTM layer (Hochreiter & Schmidhuber, 1997; Graves & Schmidhuber, 2005) in encoder withe parameters $\eta$. The function $f_{\theta,\eta}$ simulates the dynamics of the Neural-PDE with paramaters $\theta$ and $\eta$. By applying Bidirectional neural network, all grid data are transferred and it enables LSTM to simulate the PDEs as :

$$v_i^{n+1} = f_\theta\big(f_\eta(\boldsymbol{h}_{i+1}(\hat{\hat{u}}), \hat{u}_i^n, \boldsymbol{h}_{i-1}(\hat{u}))\big) \tag{3.1.11}$$

$$\boldsymbol{h}_{i+1}(\hat{u}) \equiv f_\eta(\hat{u}_{i+1}^n, \hat{u}_{i+2}^n, \hat{u}_{i+3}^n, \cdots, \hat{u}_k^n). \tag{3.1.12}$$

For a time-dependent PDE, if we map all our grid data into an input matrix which contains the information of $\delta x, \delta t$, then the neural network would regress such coefficients as constants and will learn and filter the physical rules from all the $k$ mesh grids data as:

$$v_i^{n+1} = f_{\theta,\eta}\big(\hat{u}_0^n, \hat{u}_1^n, \hat{u}_2^n, \cdots, \hat{u}_k^n\big) \tag{3.1.13}$$

The LSTM neural network is designed to overcome the vanishing gradient issue through hidden layers, therefore we use such recurrent structure to increase the stability of the numerical approach in deep learning. The highly nonlinear function $f_{\theta,\eta}$ simulates the dynamics of updating rules for $u_i^{n+1}$, which works in a way similar to a finite difference method (section 2.2) or a finite element method.

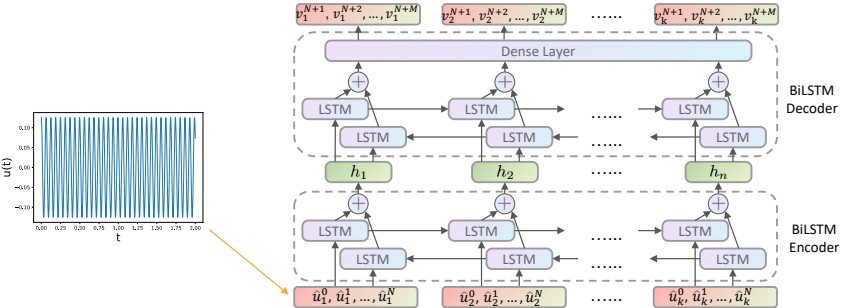

Figure 3: Neural-PDE

## 3.2 NEURAL-PDE

In particular, we use the bidirectional LSTM (Hochreiter & Schmidhuber, 1997; Graves & Schmidhuber, 2005) to better retain the state information from data on grid points which are neighbourhoods in the mesh but far away in input matrix.

The right frame of Figure 3 shows the overall design of the Neural-PDE. Denote the time series data at collocation points as $\boldsymbol{a}_1^N, \boldsymbol{a}_2^N, \cdots, \boldsymbol{a}_k^N$ with $\boldsymbol{a}_i^N = [\hat{u}_i^0, \hat{u}_i^1, \cdots, \hat{u}_i^N]$ at $i^{th}$ point. The superscript represents different time points. The Neural-PDE takes the past states $\{\boldsymbol{a}_1^N, \boldsymbol{a}_2^N, \cdots, \boldsymbol{a}_k^N\}$ of all collocation points, and outputs the predicted future states $\{\boldsymbol{b}_1^M, \boldsymbol{b}_2^M, \cdots, \boldsymbol{b}_k^M\}$, where $\boldsymbol{b}_i^M = [v_i^{N+1}, v_i^{N+2}, \cdots, v_i^{N+M}]$ is the Neural-PDE prediction for the $i^{th}$ collocation point at time points from $N+1$ to $N+M$. The data from time point 0 to $N$ are the training data set.

The Neural-PDE is an encoder-decoder style sequence model that first maps the input data to a low dimensional latent space that

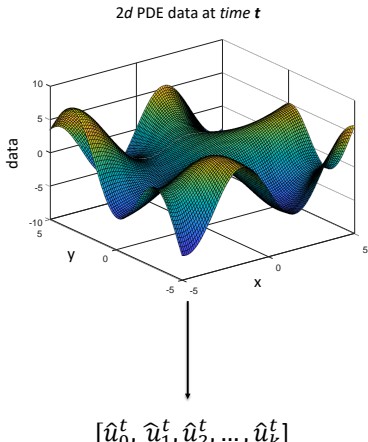

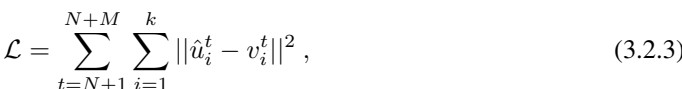

Figure 2: An example of maping $2d$ data matrix into $1d$ vector where $k = N_x \times N_y$ and $N_x$ and $N_y$ are the numbers of grid points on $x$ and $y$, respectively.

$$\boldsymbol{h}_i = \overrightarrow{\text{LSTM}}(\boldsymbol{a}_i) \oplus \overleftarrow{\text{LSTM}}(\boldsymbol{a}_i), \qquad (3.2.1)$$

where $\oplus$ denotes concatenation and $\boldsymbol{h}_i$ is the latent embedding of point $\boldsymbol{a}_i$ under the environment.

One then decoder, another bi-lstm with a dense layer:

$$v_i = \left(\overrightarrow{\text{LSTM}}(\boldsymbol{h}_i) \oplus \overleftarrow{\text{LSTM}}(\boldsymbol{h}_i)\right) \cdot \mathbf{W}, \qquad (3.2.2)$$

where $\mathbf{W}$ is the learnable weight matrix in the dense layer.

During training process, mean squared error (MSE) loss $\mathcal{L}$ is used as we typically don't know the specific form of the PDE.

$$\mathcal{L} = \sum_{t=N+1}^{N+M} \sum_{i=1}^{k} ||\hat{u}_i^t - v_i^t||^2, \qquad (3.2.3)$$

## 3.3 DATA INITIALIZATION AND GRID POINT RESHAPE

In order to feed data into our sequence model framework, we map the PDE solution data onto a $K \times N$ matrix, where $K \in \mathbb{Z}^+$ is the dimension of the grid points and $N \in \mathbb{Z}^+$ is the length of the time series data on each grid point. There is no regularization for the input order of the grid points data in the matrix because of the bi-directional structure of the Neural-PDE. For example, a $2d$

| | $1d$ **Wave** | $2d$ **Heat** | $2d$ **Burgers'** | **Fluid System** |
|---|---|---|---|---|
| **MSE** | $7.4444E - 5$ | $7.0741E - 6$ | $1.4018E - 5$ | $6.1631E - 7$ |

Table 1: Neural-PDE shows very small test MSE on 4 different PDEs.

| | **Allen–Cahn** | **Burgers** |
|---|---|---|
| **PINN** | $7.0E - 3$ | $6.7E - 4$ |
| **Neura-PDE** | $2.9E - 5$ | $2.4E - 5$ |

Table 2: Neural-PDE outperforms baseline in test MSE on $1d$ Allen-Cahn and Burgers equations.

heat equation at some time $t$ is reshaped into a $1d$ vector (See Fig. 2). Then the matrix is formed accordingly.

For a $n$-dimensional time-dependent partial differential equation with $K$ collocation points, the input and output data for $t \in (0, T)$ will be of the form:

$$\boldsymbol{A}(K, N) = \begin{bmatrix} \boldsymbol{a}_0^N \\ \vdots \\ \boldsymbol{a}_\ell^N \\ \vdots \\ \boldsymbol{a}_K^N \end{bmatrix} = \begin{bmatrix} \hat{u}_0^0 & \hat{u}_0^1 & \cdots & \hat{u}_0^n & \cdots & \hat{u}_0^N \\ \vdots & \vdots & \ddots & \vdots & \ddots & \vdots \\ \hat{u}_\ell^0 & \hat{u}_\ell^1 & \cdots & \hat{u}_\ell^n & \cdots & \hat{u}_\ell^N \\ \vdots & \vdots & \ddots & \vdots & \ddots & \vdots \\ \hat{u}_K^0 & \hat{u}_K^1 & \cdots & \hat{u}_K^n & \cdots & \hat{u}_K^N \end{bmatrix} \quad (3.3.1)$$

$$\boldsymbol{B}(K, M) = \begin{bmatrix} \boldsymbol{b}_0^M \\ \vdots \\ \boldsymbol{b}_\ell^M \\ \vdots \\ \boldsymbol{b}_K^M \end{bmatrix} = \begin{bmatrix} v_0^{N+1} & v_0^{N+2} & \cdots & v_0^{N+m} & \cdots & v_0^{N+M} \\ \vdots & \vdots & \ddots & \vdots & \ddots & \vdots \\ v_\ell^{N+1} & v_\ell^{N+2} & \cdots & v_\ell^{N+m} & \cdots & v_k^{N+M} \\ \vdots & \vdots & \ddots & \vdots & \ddots & \vdots \\ v_K^{N+1} & v_K^{N+2} & \cdots & v_K^{N+m} & \cdots & v_K^{N+M} \end{bmatrix} \quad (3.3.2)$$

Here $N = \frac{T}{\delta t}$ and each row $\ell$ represents the time series data at the $\ell^{th}$ mesh grid, and $M$ is the time length of the predicted data.

By adding Bidirectional LSTM encoder in the Neural-PDE, it will automatically extract the information from the time series data as:

$$\boldsymbol{B}(K, M) = PDESolver(\boldsymbol{A}(K, N)) = PDESolver(\boldsymbol{a}_0^N, \boldsymbol{a}_1^N, \cdots \boldsymbol{a}_i^N, \cdots, \boldsymbol{a}_K^N) \quad (3.3.3)$$

## 4 COMPUTER EXPERIMENTS

Since the Neural-PDE is a sequence to sequence learning framework which allows one to predict within any time period by the given data. One may test the Neural-PDE using different permutations of training and predicting time periods for its efficiency, robustness and accuracy. In the following examples, the whole dataset is randomly splitted in $80\%$ for traning and $20\%$ for testing. We will predict the next $t_p \in [31 \times \delta t, 40 \times \delta t]$ PDE solution by using its previous $t_{tr} \in [0, 30 \times \delta t]$ data as:

$$\boldsymbol{B}(K, 10) = PDESolver(\boldsymbol{A}(K, 30)) \quad (4.0.1)$$

Table 1 summaries the experimental results of the Neural-PDE model on 4 different PDEs, which achieve extremely small MSEs from $\sim 10^{-5}$ to $\sim 10^{-7}$. Table 2 shows the comparison results of our proposed Neural-PDE with the state-of-the-art method Physically Informed Artificial Neural Networks (PINN) (Raissi et al., 2019) on two PDEs ($1d$ Allen-Cahn and $1d$ Burgers' equation). Neural-PDE is able to outperform PINN while having much less parameters, where PINN contains 4 hidden layers with 200 neurons per layer and Neural-PDE only consists of 3 layers (2 bi-lstm with 20 neurons per layer and 1 dense output layer with 10 neurons).

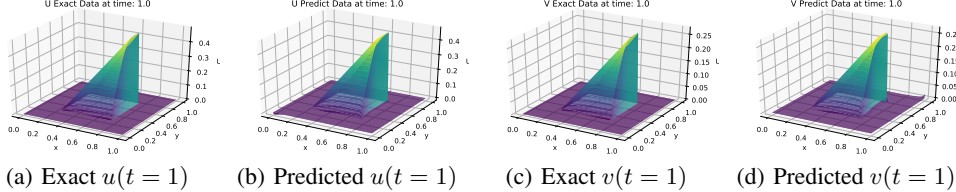

(a) Exact $u(t=1)$     (b) Predicted $u(t=1)$     (c) Exact $v(t=1)$     (d) Predicted $v(t=1)$

Figure 4: Neural-PDE shows ideal prediction on Burgers' equation.

EXAMPLE: INVISCID BURGERS' EQUATION

Inviscid Burgers' equation is a classical nonlinear PDE in fluid dynamics. In this example, we consider an invicid Burgers' equation which has the following hyperbolic form:

$$\frac{\partial u}{\partial t} + u\frac{\partial u}{\partial x} + v\frac{\partial u}{\partial y} = 0, \quad \frac{\partial v}{\partial t} + u\frac{\partial u}{\partial x} + v\frac{\partial u}{\partial y} = 0 \tag{4.0.2}$$

$$\Omega = [0,1] \times [0,1], t \in [0,1], \tag{4.0.3}$$

and with initial and boundary conditions:

$$u(0.25 \leq x \leq 0.75, \ 0.25 \leq y \leq 0.75, t=0) = 0.9 \tag{4.0.4}$$

$$v(0.25 \leq x \leq 0.75, \ 0.25 \leq y \leq 0.75, t=0) = 0.5 \tag{4.0.5}$$

$$u(0,y,t) = u(1,y,t) = v(x,0,t) = v(x,1,t) = 0 \tag{4.0.6}$$

The invicid Burgers' equation is hard to deal with in numerical PDEs due to the discontinuities (shock waves) in the solutions. We use a upwinding finite difference scheme to create the training data and put the velocity $u,v$ in to the input matrix. Let $\delta x = \delta y = 10^{-2}, \delta t = 10^{-3}$, our empirical results (see Figure 4) show that the Neural-PDE is able to learn the shock waves, boundary conditions and the rules of the equation, and predict $u$ and $v$ simultaneously with an overall MSE of $1.4018 \times 10^{-5}$. The heat maps of exact solution and predicted solution are shown in Figure 5.

EXAMPLE: MULTISCALE MODELING: COUPLED CAHN–HILLIARD–NAVIER–STOKES SYSTEM

Finally, let's consider the following $2d$ Cahn–Hilliard–Navier–Stokes system widely used for modeling complex fluids:

$$\boldsymbol{u}_t + \boldsymbol{u} \cdot \nabla \boldsymbol{u} = -\nabla p + \nu \Delta \boldsymbol{u} - \phi \nabla \mu \,, \tag{4.0.7}$$

$$\phi_t + \nabla \cdot (\boldsymbol{u}\phi) = M\Delta\mu \,, \tag{4.0.8}$$

$$\mu = \lambda(-\Delta\phi + \frac{\phi}{\eta^2}(\phi^2 - 1)) \tag{4.0.9}$$

$$\nabla \cdot \boldsymbol{u} = 0 \tag{4.0.10}$$

In this complicated example we will use the following initial condition:

$$\phi(x,y,0) = (\frac{1}{2} - 50\tanh(f_1 - 0.1)) + (\frac{1}{2} - 50\tanh(f_2 - 0.1)), \ I.C. \tag{4.0.11}$$

$$f_1 = \sqrt{(x+0.12)^2 + (y)^2}, \ f_2 = \sqrt{(x-0.12)^2 + (y)^2} \tag{4.0.12}$$

$$\text{with } x \in [-0.5, 0.5], \ y \in [-0.5, 0.5], \ t \in [0,1], \ M = 0.1, \ \nu = 0.01 \tag{4.0.13}$$

This fluid system can be derived by the energetic variational approach (Forster, 2013). The complex fluids system has the following features: the micro-structures such as the molecular configurations, the interaction between different scales and the competition between multi-phase fluids (Hyon et al., 2010). Here $\boldsymbol{u}$ is the velocity and $\phi(x,y,t) \in [0,1]$ denotes the volume fraction of one fluid phase. $M$ is the diffusion coefficient and $\mu$ is the chemical potential of $\phi$. Equation (4.0.10) indicates the incompressibility of the fluid. Solving such PDE system is notorious because of its high nonlinearity and multi-physical and coupled features. One may use the decoupled projection method (Guermond et al., 2006) to numerically solve it efficiently or an implicit method which however is computationally

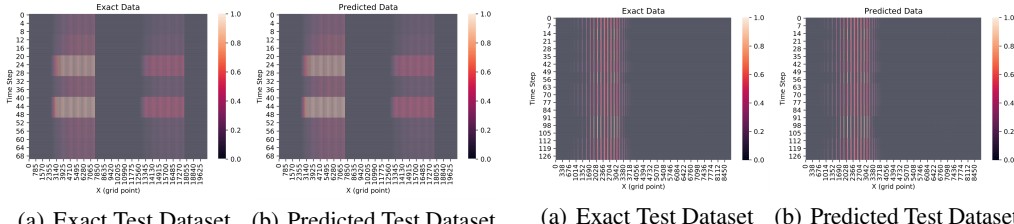

(a) Exact Test Dataset    (b) Predicted Test Dataset    (a) Exact Test Dataset    (b) Predicted Test Dataset

Figure 5: Neural-PDE shows ideal prediction on $2d$ Burgers Equation.

Figure 6: Neural-PDE shows ideal prediction on Fluid System.

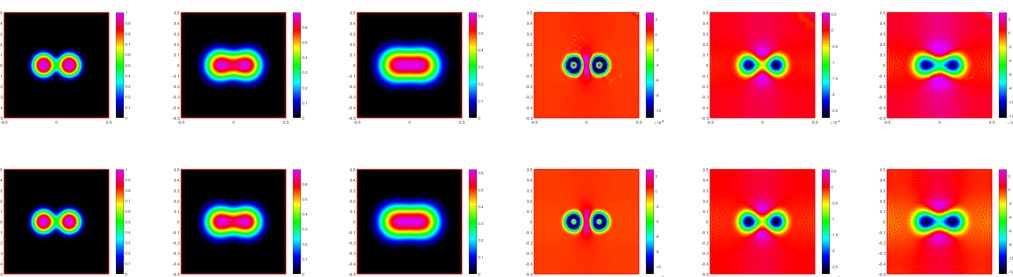

Figure 7: Predicted data by Neural-PDE (first row) and the exact data (second row) of volume fraction $\phi$ (column 1-3) and pressure $p$ (column 4-6). The graphs of each columns 1-3 and 4-6 represent the time states of $t_1, t_2, t_3$, where $0 \leq t_1 < t_2 < t_3 \leq 1$.

expensive. Another challenge of deep learning in solving a system like this is how to process the data to improve the learning efficiency when the input matrix consists of variables such as $\phi \in [0, 1]$ with large magnitude value and variable of very small values such as $p \sim 10^{-5}$. For the Neural-PDE to better extract and learn the physical features of variables in different scales, we normalized the $p$ data with a $sigmoid$ function. Set $\delta t = 5 \times 10^{-4}$, and here the training dataset is generated by a FEM solver FreeFem++ (Hecht, 2012) using a Crank-Nicolson finite element scheme. Our Neural-PDE prediction shows that the physical features of $p$ and $\phi$ have been successfully captured with an overall MSE: $6.1631 \times 10^{-7}$ (see Figure 7).

## 5    CONCLUSIONS

In this paper, we proposed a novel sequence recurrent deep learning framework: Neural-PDE, which is capable of intelligently filtering and learning solutions of time-dependent PDEs. One key innovation of our method is that the time marching method from the numerical PDEs is applied in the deep learning framework, and the neural network is trained to explore the accurate numerical solutions for prediction.

The state-of-the-art researches have shown the promising power of deep learning in solving high-dimensional nonlinear problems in engineering, biology and finance with efficiency in computation and accuracy in prediction. However, there are still unresolved issues in applying deep learning in PDEs. For instance, the stability and convergence of the numerical algorithms have been rigorously studied by applied mathematicians. Due to the high nonlinearity of the neural network system and the curse of dimensionality (Hutzenthaler et al., 2019), theorems guiding stability and convergence of solutions predicted by the neural network are to be revealed.

Lastly, it would be helpful and interesting if one can theoretically characterize a numerical scheme from the neural network coefficients and learn the forms or mechanics from the scheme and prediction. We leave these questions for further study.

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

# A APPENDIX

## A.1 FINITE ELEMENT METHOD

Finite element method (FEM) is a powerful numerical method in solving PDEs. Consider a 1D wave equation of $u(x,t)$:

$$u_{tt} - v^2 u_{xx} = f, \quad x \in [a,b] \equiv \Omega \subset \mathbb{R}, \quad t \in \mathbb{R}^+ \cup \{0\} , \tag{A.1.1}$$
$$u_x(a,t) = u_x(b,t) = 0 . \tag{A.1.2}$$

The function $u$ is approximated by a function $u_h$ :

$$u(x,t) \approx u_h(x,t) = \sum_{i=1}^{N} a_i(t)\psi_i(x) \tag{A.1.3}$$

$$\tag{A.1.4}$$

where $\psi_i \in V$, is the basis functions of some FEM space $V$, and $a_i^n$ denotes the coefficients. $N$ denotes the degrees of freedom.

Multiply the equation with an arbitrary test function $\psi_j$ and integral over the whole domain we have:

$$\int_\Omega u_{tt}\psi_j \, dx + v^2 \int_\Omega \nabla u \nabla \psi_j \, dx = \int_\Omega f\psi_j \, dx \tag{A.1.5}$$

$$\tag{A.1.6}$$

and approximate $u(x,t)$ by $u_h$:

$$\sum_{i}^{N} \frac{\partial^2 a_i(t)}{\partial t^2} \underbrace{\int_\Omega \psi_i\psi_j \, dx}_{\mathbf{M}_{i,j}} + v^2 \sum_{i}^{N} a_i(t) \underbrace{\int_\Omega \nabla\psi_i\nabla\psi_j \, dx}_{\mathbf{A}_{i,j}} = \underbrace{\int_\Omega f\psi_j \, dx}_{\mathbf{b}} , \tag{A.1.7}$$

$$\equiv \mathbf{M}^T \mathbf{a}_{tt} + v^2 \mathbf{A}^T \mathbf{a} = \mathbf{b} . \tag{A.1.8}$$

Here $\mathbf{M}$ is the mass matrix and $\mathbf{A}$ is the stiffness matrix, $\mathbf{a} = (a_1, .., a_N)^t$ is a $N \times 1$ vector of the coefficients at time $t$. The central difference method for time discretization indicates that (Johnson, 2012):

$$\mathbf{a}^{n+1} = 2\mathbf{a}^n - \mathbf{a}^{n-1} + \mathbf{M}^{-1}(\mathbf{b} - v^2\mathbf{A}^T\mathbf{a}^n) , \tag{A.1.9}$$

$$u^{n+1} \approx u_h^{n+1} = \sum_{i}^{N} a_i^{n+1}\psi_i(x) . \tag{A.1.10}$$

## A.2 LONG SHORT-TERM MEMORY

Long Short-Term Memory Networks (LSTM) (Hochreiter & Schmidhuber, 1997; Graves & Schmidhuber, 2005) are a class of artificial recurrent neural network (RNN) architecture that is commonly used for processing sequence data and can overcome the gradient vanishing issue in RNN. Similar to most RNNs (Mikolov et al., 2011), LSTM takes a sequence $\{\boldsymbol{x}_1, \boldsymbol{x}_2, \cdots, \boldsymbol{x}_t\}$ as input and learns hidden vectors $\{\boldsymbol{h}_1, \boldsymbol{h}_2, \cdots, \boldsymbol{h}_t\}$ for each corresponding input. In order to better retain long distance information, LSTM cells are specifically designed to update the hidden vectors. The computation process of the forward pass for each LSTM cell is defined as follows:

$$\boldsymbol{i}_t = \sigma(\mathbf{W}_i^{(x)}\boldsymbol{x}_t + \mathbf{W}_i^{(h)}\boldsymbol{h}_{t-1} + \mathbf{W}_i^{(c)}\boldsymbol{c}_{t-1} + \boldsymbol{b}_i) ,$$
$$\boldsymbol{f}_t = \sigma(\mathbf{W}_f^{(x)}\boldsymbol{x}_t + \mathbf{W}_f^{(h)}\boldsymbol{h}_{t-1} + \mathbf{W}_f^{(c)}\boldsymbol{c}_{t-1} + \boldsymbol{b}_f) ,$$
$$\boldsymbol{c}_t = \boldsymbol{f}_t\boldsymbol{c}_{t-1} + \boldsymbol{i}_t \tanh(\mathbf{W}_c^{(x)}\boldsymbol{x}_t + \mathbf{W}_c^{(h)}\boldsymbol{h}_{t-1} + \boldsymbol{b}_c) ,$$
$$\boldsymbol{o}_t = \sigma(\mathbf{W}_o^{(x)}\boldsymbol{x}_t + \mathbf{W}_o^{(h)}\boldsymbol{h}_{t-1} + \mathbf{W}_o^{(c)}\boldsymbol{c}_t + \boldsymbol{b}_o),$$
$$\boldsymbol{h}_t = \boldsymbol{o}_t \tanh(\boldsymbol{c}_t) ,$$

where $\sigma$ is the logistic sigmoid function, $\mathbf{W}$s are weight matrices, $\boldsymbol{b}$s are bias vectors, and subscripts $\boldsymbol{i}, \boldsymbol{f}, \boldsymbol{o}$ and $\boldsymbol{c}$ denote the input gate, forget gate, output gate and cell vectors respectively, all of which have the same size as hidden vector $\boldsymbol{h}$.

This LSTM structure is used in the paper to simulate the numerical solutions of partial differential equations.

### A.3 EXAMPLES

#### A.3.1 WAVE EQUATION

Consider the $1d$ wave equation:

$$u_{tt} = cu_{xx}, \ x \in [0,1], \ t \in [0,2] \,, \tag{A.3.1}$$

$$u(x,0) = sin(4\pi x) \tag{A.3.2}$$

$$u(0,t) = u(1,t) \tag{A.3.3}$$

Let $c = \frac{1}{16\pi^2}$ and use the analytical solution given by the characteristics for the training and testing data:

$$u(x,t) = \frac{1}{2}(sin(4\pi x + t) + sin(4\pi x - t)) \tag{A.3.4}$$

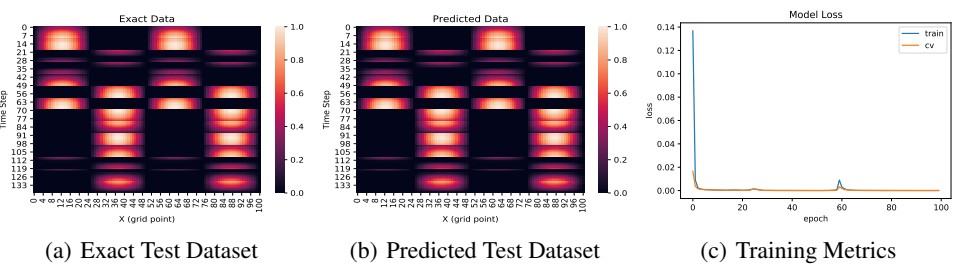

| (a) Exact Test Dataset | (b) Predicted Test Dataset | (c) Training Metrics |

Figure 8: $\delta x = 1 \times 10^{-2}$, $\delta t = 1 \times 10^{-3}$, MSE: $7.4444 \times 10^{-5}$, the whole time period length is $n_t = 2000$ and the mesh grid size is 101, the test dataset size is 14 and thus the total discrete testing time period is 140, figure $(a)$ and figure$(b)$ are the heat map for the exact test data and our predicted test data. Figire$(c)$ shows both training and cross-validation errors of Neural-PDE convergent within 10 epochs.

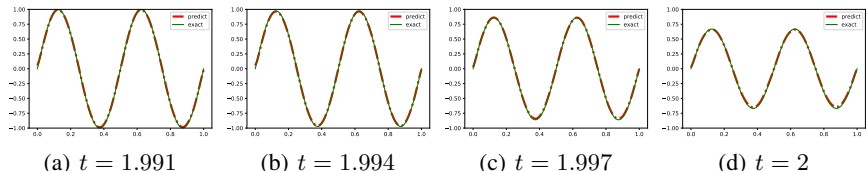

| (a) $t = 1.991$ | (b) $t = 1.994$ | (c) $t = 1.997$ | (d) $t = 2$ |

Figure 9: We selected the final states for computation and the results indicate that Neural-PDE is robust in capturing the physical laws of wave equation and predicting the sequence time period.

#### A.3.2 HEAT EQUATION

The heat equation describes how the motion or diffusion of a heat flow evolves over time. The Black–Scholes model (Black & Scholes, 1973) is also developed based on the physical laws behind the heat equation. Rather than the 1D case that maps the data into a matrix (**??**) with its original spatial locations, the high dimensional PDEs grids are mapped into matrix without regularization of the position, and the experimental results show that Neural-PDE is able to capture the valuable features regardless of the order of the mesh grids in the matrix. Let's start with a 2D heat equation as follows:

$$u_t = u_{xx} + u_{yy}, \tag{A.3.5}$$

$$u(x,y,0) = \begin{cases} 0.9, & \text{if } (x-1)^2 + (y-1)^2 < 0.25 \\ 0.1, & \text{otherwise} \end{cases} \tag{A.3.6}$$

$$\Omega = [0,2] \times [0,2], \ t \in [0,0.15] \tag{A.3.7}$$

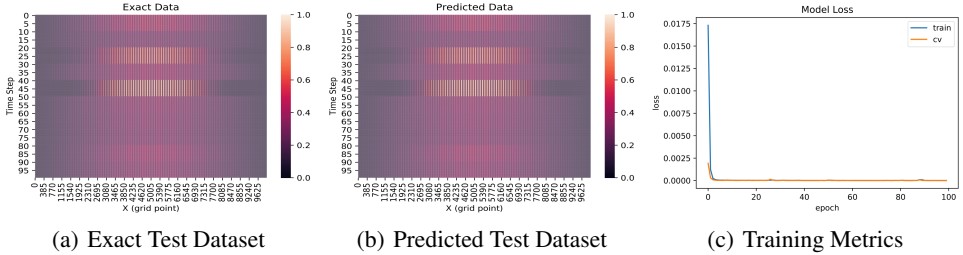

(a) Exact Test Dataset      (b) Predicted Test Dataset      (c) Training Metrics

Figure 10: $\delta x = 0.02, \delta y = 0.02, \delta t = 10^{-4}$, MSE: $7.0741 \times 10^{-6}$, the size of the test data is 10 and the test time period is 140.

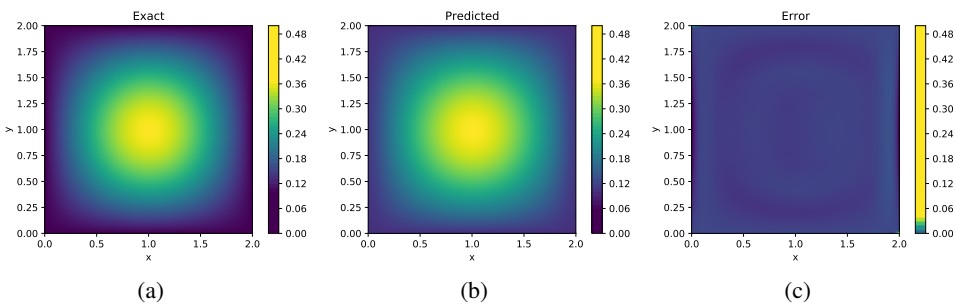

(a)            (b)            (c)

Figure 11: figure $(a)$ is the exact solution $u(x, y, t = 0.15)$ at the final state and figure $(b)$ is the model's prediction. Figure $(c)$ is the error map.

