# OpenReview forum: "Neural Time-Dependent Partial Differential Equation"
_ICLR.cc/2021/Conference — Reject_

### Official Review · AnonReviewer4 · 2020-10-22
**Need the comparison with Conv-LSTM**

**Rating:** 3
**Confidence:** 4

**Review:**

Summary:
The paper proposes to use LSTM to learn partial differential equations. Experiments include Wave equation, Heat equation, Burgers' equation, and Navier-Stokes equation. They compare with PINN on Allen-Cahn and Burgers equation.

The paper is nicely written and easy to understand. But I have several concerns:

Novelty:
Using the RNN/LSTM type of networks for time series / time-dependent PDEs doesn't seem to be novel. Convolution-LSTM has been widely used in these tasks e.g. (https://arxiv.org/abs/2002.03014).
Instead of using Convolution, the author uses the seq-to-seq structure. Unfortunately, there is no sufficient justification or any empirical comparison. Indeed, the seq-to-seq loses the location information compared to convolution-LSTM. I doubt if it can outperform the convolution.

Experiments:
The work compares again PINN. In my opinion, it is not a fair comparison. If I understand correctly, the Neural-PDE proposed in this paper uses the ground truth data $u$ for training, but PINN doesn't have the access to these ground truth data. To have a fair comparison, I suggest comparing with LSTM type solvers, for example, Conv-LSTM, Unet-RNN, TF-net (https://arxiv.org/abs/1911.08655)

Error Metrics:
The paper uses MSE as the loss metric. It's no problem to train with MSE, but when reporting the error rate, it's better to use the relative-L2 error, or at least some relative error. The absolute MSE error looks very encouraging, but it has no physical meaning, in my opinion.

Therefore, I am sorry to suggest rejection.

---

### Official Review · AnonReviewer2 · 2020-10-26
**DNN-based dynamical systems' solutions prediction.**

**Rating:** 5
**Confidence:** 5

**Review:**

This paper is about using deep neural nets to predict solutions of dynamical systems described by PDEs. The paper is overall well written, and the contribution is clearly stated.

However, I do have several issues with the general principle and what is actually achievable with such data-driven approach to solving dynamical systems. Indeed, the idea is as simple as correct; mainly using a DNN with dynamics, e.g., RNNs, to predict solutions of a dynamical systems over time, based on some past (timewise) training data. This is clearly stated in the paper and there is nothing wrong with this idea, except that it seems to me that the authors are exaggerating the value of their findings.

For instance, when the authors claim that they are learning the law of physics of the PDE, as well as its boundary conditions, they  cannot be further from the truth. Indeed, fitting a predictor based on past data is not a proof of learning anything structural about the PDE, since what you are learning is simply one regime of the solutions, based on the available data. In other words, if you use data points from your transient regime and try to predict the solutions over a time horizon further way from your transient regime, your predictor will fail, simply because you don't have anything in your data to inform the predictor (your DNN) about he future regime ! I hope this makes sense. As a follow up on this point, I want to know in all your examples, what is the time interval over which the learning data is collected ? is it close to the test time instants ? I would expect it to be rather close. Similarly, in your fluid dynamics example, i.e., coupled NS equation, can you please show the results for a more turbulent case, .e.g., $\mu=1e-3$, for which it is rather impossible to predict the steady state from transient points ! These tests will show you that you are not learning the physics of the system, as you are claiming throughout the paper; please remove these exaggerated statements.

---

### Official Review · AnonReviewer1 · 2020-10-27
**Learning PDEs with LSTMs**

**Rating:** 4
**Confidence:** 4

**Review:**

This work proposes a sequence-to-sequence approach for learning the time evolution of PDEs. The method employs a bi-directional LSTM to predict solutions of a PDE-based formulation for a chosen number of time steps. By itself this is an interesting, and important goal, but the method does not seem to contain any novel components apart from demonstrating that LSTMs can be used to learn data from PDEs. The paper only compares to a simple form of PINNs, but not to a variety of other time forecasting algorithms available in the deep learning field (LSTM are just one of many methods used these days, a more state of the art one being e.g. transformers). In addition, the examples only contain single cases with relatively simple model equations.

In terms of exposition, the paper spends a significant amount of space discussing relatively basic finite difference, finite element, and LSTM schemes.

Most importantly, from an ICLR paper, I would expect advances in terms of learning methodology: here the authors use a regular bi-directional LSTM, with a fully supervised loss formulation. No changes or modifications are proposed to improve learning the content of PDE or coupling the PDE formulations with the LSTM. Hence, unfortunately, the submission does not really go beyond a demonstration that LSTMs can learn the PDEs under consideration. While this is good to see, I don't think it's a sufficiently large step forward for recommending acceptance as an ICLR poster.

---

### Official Review · AnonReviewer3 · 2020-10-28
**Decent paper. Example problems are too simple. A few more successful baselines could had been considered**

**Rating:** 5
**Confidence:** 5

**Review:**

The paper is a good contribution to spatio-temporal modeling in complex physics. However, the novelty of the paper is unclear. It is an extended version of Neural-ODEs but used a Bi-LSTM.

Pros: A good new approach but very similar to Neural-ODE. There is nothing particularly interesting in this approach.
Cons: Comparison metrics do not show a major difference. $O(10^{-4})$ and $O(10^{-5})$ as RMSE are essentially equal good for a super simple example like inviscid Burgers or any of the other examples.

1) There is a long line of literature that looks at spatio-temporal prediction of chaotic dynamical systems, systems that are more complex than the ones considered in this paper with machine learning and shows state-of-the-art performance with reservoir computers. They do outperform MLP, CNN, and other ML architectures and are much easier to train. **https://journals.aps.org/prl/abstract/10.1103/PhysRevLett.120.024102; **https://npg.copernicus.org/articles/27/373/2020/; **https://www.sciencedirect.com/science/article/abs/pii/S0893608020300708; **https://agupubs.onlinelibrary.wiley.com/doi/10.1029/2020GL087776. The authors should cite these papers and also consider RC as a baseline.

2) I have some concerns with the comparison with PINNs. There is an advantage to using PINNs that is overlooked by the authors. Since they optimize the residual of the PDEs, they can work without any data at all and can also satisfy BCs by construction. I was wondering how many data points the authors have considered during training PINNs and comparing with Neural-PDEs.

3) Figure 4's quality is very poor. For an average reader, it is impossible for me to make out the axis labels etc.

4) PINNs can be used to satsify BCs by construction. That would undermine one of the points of this paper (satisfying BCs better than PINNs) . I would like to see a plot that shows the how well the BCs are satisfied during inference as we move in time.

5) I would like to see one more test case that is a tad-bit more challenging than the ones considered in this problem. Something chaotic, like Lorenz 96 system can be used. The true potential of the architecture would be revealed when used on a system like that.
6) Figure 5,6,7 quality is very low. The heat equation considered is too simple for a spatio-temporal prediction problem. There is hardly any major variation with time to pose a challenge. Something simple like the Von-Karman vortex sheet problem in fluid dynamics would be a better test case. The authors should consider looking at test cases (at least a few) to pose a challenge for their architecture

7) I agree with the authors that there is a difference in RMSE when compared to PINNs . PINNs are $\sim O(10^{-4})$ and their framework is $\sim O(10^{-5})$. But this just means that both the frameworks are doing amazingly well. The difference would hardly be noticeably, let along have any physical significance for problems such as these. However, this same difference on a very chaotic or turbulent flow would make a difference when it comes to analyzing the long terms statistics. The authors should look at the state-of-the-art of PDE solving with ML and relevant literature that looked at meaningful fluid dynamics/computational physics problems and consider at least one test case that is challenging enough. Example papers: https://arxiv.org/abs/1808.02983, https://arxiv.org/abs/2002.00021

---

### Decision · Program_Chairs · 2021-01-07
**Final Decision**

**Decision:**

Reject

**Comment:**

The paper introduces an approach for learning the dynamics of PDEs. It makes use of bi-directional LSTMs trained to regress future values from past observations, up to a given horizon. Experiments are performed on data generated from numerical solvers on two examples, inviscid Burgers and a Navier-Stokes system. While the topic is fine, the solution is nothing more than regression with sequence models and only shows that RNNs could learn to predict the data generated by these PDEs. The reviewers also highlight that the comparison with the baselines is not appropriate.